# A Gene Expression High-Throughput Screen (GE-HTS) for Coordinated Detection of Functionally Similar Effectors in Cancer

**DOI:** 10.3390/cancers12113143

**Published:** 2020-10-27

**Authors:** Chaitra Rao, Dianna H. Huisman, Heidi M. Vieira, Danielle E. Frodyma, Beth K. Neilsen, Binita Chakraborty, Suzie K. Hight, Michael A. White, Kurt W. Fisher, Robert E. Lewis

**Affiliations:** 1Eppley Institute, Fred & Pamela Buffett Cancer Center, University of Nebraska Medical Center, Omaha, NE 68198, USA; chaitra.rao@unmc.edu (C.R.); dianna.huisman@unmc.edu (D.H.H.); heidi.vieira@unmc.edu (H.M.V.); danielle.frodyma@unmc.edu (D.E.F.); beth.neilsen@unmc.edu (B.K.N.); 2Pharmacology and Cancer Biology, Duke University Medical Center, Durham, NC 27710, USA; binita.das@duke.edu; 3Moores Cancer Center, University of California San Diego, La Jolla, CA 92037, USA; shight@health.ucsd.edu; 4Chief Scientific Officer, Samumed, LLC, San Diego, CA 92121, USA; michaelw@samumed.com; 5Department of Pathology and Microbiology, University of Nebraska Medical Center, Omaha, NE 68198, USA; kfisher@unmc.edu

**Keywords:** high-throughput screens, functional signature ontology, cancer susceptibility genes, Ras-driven cancer

## Abstract

**Simple Summary:**

Approximately 30% of human cancer patients carry active Ras mutations, which play a central role in cancer cell malignancy. Although Ras serves as an excellent drug target, significant challenges still exist to directly inhibit all forms of oncogenic Ras. Identifying functionally similar proteins with known inhibitors may overcome this bottleneck in Ras-driven cancers. In this review, we investigate the use of a functional genome-scale screen termed Functional Signature Ontology (FUSION) to screen small interfering RNAs (siRNA), microRNA mimics, and small molecules or complex mixtures of natural products for coordinated detection of new targets and small molecules for cancer therapy. After computational filtering and prioritizing of targets identified in FUSION, biological validation was performed to elucidate their mechanism of action in cancer and to provide an initial evaluation of their potential to serve as therapeutic targets. We also discuss the guidelines for design, optimization, and analysis to increase the applicability and generalizability of future FUSION screens.

**Abstract:**

Genome-wide, loss-of-function screening can be used to identify novel vulnerabilities upon which specific tumor cells depend for survival. Functional Signature Ontology (FUSION) is a gene expression-based high-throughput screening (GE-HTS) method that allows researchers to identify functionally similar proteins, small molecules, and microRNA mimics, revealing novel therapeutic targets. FUSION uses cell-based high-throughput screening and computational analysis to match gene expression signatures produced by natural products to those produced by small interfering RNA (siRNA) and synthetic microRNA libraries to identify putative protein targets and mechanisms of action (MoA) for several previously undescribed natural products. We have used FUSION to screen for functional analogues to Kinase suppressor of Ras 1 (KSR1), a scaffold protein downstream of Ras in the Raf-MEK-ERK kinase cascade, and biologically validated several proteins with functional similarity to KSR1. FUSION incorporates bioinformatics analysis that may offer higher resolution of the endpoint readout than other screens which utilize Boolean outputs regarding a single pathway activation (i.e., synthetic lethal and cell proliferation). Challenges associated with FUSION and other high-content genome-wide screens include variation, batch effects, and controlling for potential off-target effects. In this review, we discuss the efficacy of FUSION to identify novel inhibitors and oncogene-induced changes that may be cancer cell-specific as well as several potential pitfalls within FUSION and best practices to avoid them.

## 1. Introduction

Many cancers lack specific, durable, and effective treatment options. Although great strides have been made in cancer research, including novel therapies and detection methods, cancer continues to be a highly lethal disease [1,2]. To discover novel therapeutic targets, researchers have employed a variety of exciting new approaches for identifying and validating genes critical to tumor development and survival [3,4,5]. The use of high-throughput screening allows for the rapid interrogation of cellular targets using RNA interference (RNAi) or small molecules to identify proteins and cellular processes critical to cancer cells [6,7,8,9,10,11]. The complexity and design of the screening strategy are critical determinants to the number and confidence of possible discoveries generated from the initial screen. Screens carried out using one biological endpoint, such as a viability assay, often yield many candidate hits but often lack the specificity needed to advance to more critical discoveries. Unlike a single-endpoint assay, high-content assays can generate large amounts of complex data that have the ability to lead to detailed discoveries, but the generation and analysis of data increases exponentially. In this review, we describe the use of a Gene Expression High-Throughput Screening (GE-HTS) technique termed Functional Signature Ontology (FUSION) [12,13,14,15,16,17,18], which offers a methodology for identifying genes critical to tumor cell survival and growth and inhibitors of these processes. FUSION has been used to screen for functional similarity between proteins using 14,355 unique siRNA pools, 344 microRNA mimics, and approximately 1200 natural products fractions in a K-Ras mutant colon cancer cell line and was used to identify novel proteins required for modulation of autophagy and proteins functionally similar to Kinase suppressor of Ras 1 (KSR1) and to determine the target and mechanism of action (MoA) of uncharacterized naturally occurring small molecules [12,13,14,15,16,17,18,19,20].

## 2. Gene Expression-Based High-Throughput Screens (GE-HTS) Using Chemical Libraries

High-throughput screening (HTS) provides an unbiased approach for biological discovery using a variety of reagents to perturb cells and to measure cell-based phenotypes. Boolean or single-endpoint phenotype-based screens identify biological changes in cell state, such as induction of cell death or reduced viability, as an endpoint assay. Although these screens identify many candidate targets [21,22], they usually require multiple rounds of validation to achieve sufficient resolution of the candidate genes, small molecules, or perturbants. Moreover, determining the MoA of the identified drug compound can be challenging.

To bypass the difficulties of using single-endpoint biological phenotypes in large-scale drug screens, Gene Expression-Based High-Throughput Screening (GE-HTS) was devised to leverage a multi-panel gene expression signature as a proxy for different cellular states [6]. GE-HTS uses differential expression of a small number of genes representing the biological state of interest and then collects gene expression signatures for measurement following genetic or chemical perturbations. In principle, GE-HTS technology could be adapted to identify the modulators of any cellular process by defining the signature of the cellular process and by identifying other genes that share that signature. Also, unlike many traditional assay techniques, GE-HTS does not require significant assay customization because gene expression signature definition, amplification, and detection can be generic. The reporter genes that constitute the expression profile need not have a defined biological relationship to the reference protein; they need only to respond acutely and consistently to the reference perturbation. Stegmaier et al. pioneered this approach using a five-gene differential expression pattern (i.e., gene signature) as a surrogate phenotype to identify inducers of differentiation of myeloid leukemia [6]. Mass spectrometry of the amplified PCR products of the signature genes from the leukemia cells was used as the detection method in their study. Over 1700 chemical compounds were screened for their ability to induce the mature myeloid state. Out of 13 candidate chemical compounds identified in the screen, eight were validated by three separate biological assays and were shown to induce myeloid differentiation; however, this approach was labor-intensive and expensive [6]. The same technical approach was used to identify platelet derived growth factor receptor (PDGFR) inhibitors [23]. An improved technical approach used ligation-mediated amplification with Luminex optically addressed and barcoded microspheres with flow cytometry detection (LMF) to identify HSP90 inhibitors in prostate cancer cells using an androgen receptor inhibition signature [24]. These studies showed that GE-HTS using signatures that varied between 2 and 27 mRNA signatures could detect novel small molecules with defined functions but were limited in use to only chemical libraries under 2500 compounds. The production of libraries containing individually arrayed and pooled siRNAs targeting known genes in the human genome enabling genome-wide loss-of-function screens have been shown to elucidate novel protein biology, was an important step for the expansion of GE-HTS, and was incorporated in FUSION (discussed below).

## 3. Screening Strategies to Identify Functional Similar Proteins in Ras-Driven Cancers

Integration of results from RNAi screens with molecular profiling data enabled identification of small molecule inhibitors of Ras signaling [25,26,27,28,29]. Ras is identified to be a major driver in one-third of human cancers [30]. However, except for a few recent successes in targeting RasG12C pharmacologically [31,32], this protein has been deemed “undruggable”. Efforts have been made to identify novel effectors upstream and downstream of oncogenic Ras that can be used as an alternative approach to directly target Ras.

### 3.1. Identifying Vulnerabilities in Ras-Driven Cancers Using RNAi or CRISPR Screening

Functionally similar proteins can be identified following cellular perturbations of known genes using RNAi or CRISPR-Cas9 technology. This type of screening offers an unbiased approach for identification of potential therapeutic targets using a gene expression signature as a proxy for the phenotype of interest. Functional genomic screens are commonly conducted in two general formats: arrayed or pooled format. In the arrayed format, reagents (siRNA, shRNA, and CRISPR/Cas9) targeting a single gene are added to individual wells on a multi-well plate, which makes it ideal for screens where a qualitative or quantitative phenotypic output is desired for each individual perturbation. The final output is evaluated using a wide range of techniques such as high-content microscopy, luminescence or fluorescence, or other reporter assays. In pooled approaches, an entire RNAi or sgRNA library is delivered to a single dish of cells. This assay is typically performed for a longer period of time, which allows selection for or against transduced cells that gained or lost evolutionary advantages within the specified experimental conditions. The cells can then be examined for the presence of shRNA/sgRNA barcodes through sequencing, providing the measurement of enrichment or depletion of specific clones in response to selection. The selection of screening methods also depends on the type of genetic perturbation used. Pooled screens can be performed by two methods: positive and negative selection screens. Positive screens identify cell enrichment in response to a drug that is toxic to the starting population of cells in the form of growth assays. This selection is typically performed using pooled libraries, where the shRNA/sgRNA is depleted and only a small number of cells confer resistance to a drug, survive, and proliferate. Negative screens rely on dropouts of shRNA/sgRNA sequences from the population to identify gene perturbations that affect the cellular phenotype, such as cell death. Negative screening assays can be used to effectively target cancer cells that harbor known oncogenic mutations, without affecting the normal counterparts, resulting in identification of potential therapeutic targets [33]. CRISPR and shRNA can be used in both pooled and arrayed screens, while siRNA can be used only in arrayed screens. Both approaches require expertise in molecular biology, tissue culture, and bioinformatics, though pooled screens are comparatively less labor-intensive and inexpensive compared to an arrayed approach, which typically requires special facilities that use liquid handlers and automated workstations [34].

Multiple studies using RNAi (siRNA/shRNA) libraries targeting over 1000 genes in isogenic wild-type and mutant K-Ras cell lines [25,26] or different mutant K-Ras cell lines [27,28,29,35] identified genes selectively dependent on K-Ras function as potential therapeutic targets for cancers exhibiting aberrant K-Ras activity. Overall, these efforts have led to the identification of a wide array of proteins that are dependent on mutant K-Ras function and are classified into several cellular processes: cell survival (Bcl-XL1 and STK33), transcriptional programs (GATA2), and growth and survival signals (TAK1 and GEF-H1) [26,27,28,29,35]. Although these various studies independently found targets for Ras-driven cancers, the apparent observation was that the results had very little overlap with each other, suggesting that these findings are not universal, as these studies claim, but are more likely context-dependent. The lack of overlap between the screen results may be due to the technical aspects by which each screen was performed or to the different genetic makeup of the cell lines used [30,36]. 

The development of CRISPR/Cas9-mediated recombination into a tool for manipulation of specified genomic targets has enabled the generation of libraries containing genome-wide coverage of gene targeting that is relatively easy and cost-effective [37,38]. The feasibility of completing a genome-wide loss-of-function screen via CRISPR/Cas9 was demonstrated first by two studies [5,38]. Shalem et al. used a pooled library of 64,751 sgRNAs targeting 18,080 genes (Genome-scale CRISPR Knock-Out (GeCKO)) to examine resistance to the Rafinhibitor, Vemurafenib, in melanoma cells and ultimately identified NF1 and MED12 as top-ranking genes implicated in drug resistance [5]. Wang et al. generated a library consisting of over 73,000 sgRNAs to screen for gene knockouts that confer resistance to the purine analogue 6-thioguanine in two human cell lines [38]. A pooled CRISPR/Cas9 screen using an isogenic pair of HCT116 cell lines with and without K-Ras mutation showed that tumors with oncogenic K-Ras were highly dependent on mitochondrial oxidative phosphorylation, and inactivating metabolic enzymes such as NAD kinase (NADK) and ketohexokinase (KHK) led to reduced growth of cancer cells by 50% [39]. In another example, a recent study used the GeCKO library to identify synthetic lethal interactions between asparaginase and Wnt signaling in asparaginase-resistant T-cell acute leukemia cell lines treated with either vehicle control or a high dose of asparaginase [4]. Over the years, the versatility of the CRISPR/Cas9 system has increased, with the ability to modify Cas9 to either activate or inhibit transcription of the gene of interest (CRISPRa/CRISPRi). CRISPRi uses a catalytically dead Cas9 protein, and unlike gene knockout, suppression of the genes is nonpermanent and produces a phenotype that is more similar to RNAi studies. This system may be more effective for studying the phenotypic effects of knockdown rather than complete loss-of-function.

### 3.2. Identifying Novel Functional Protein Similarities in Ras-Driven Cancers Using FUSION

The first FUSION publication validated the use of the updated GE-HTS methodology to successfully predict and validate novel modifiers of autophagy and to identify kinase targets of uncharacterized natural products. FUSION quantitates changes in six mRNA signature (BNIP3, ALDOC, LOXL2, ACSL5, and BNIP3L) normalized to the geometric mean of two housekeeping mRNAs (PPIB and HPRT) to interrogate the expression signatures after a variety of perturbants. The screen was performed semi-robotically in the Ras-mutant human colon cell line, HCT116, and the gene expression profile for each perturbant was determined from the average of triplicate transfections in 384-well plates. In contrast to initial GE-HTS studies performed by Stegmaier et al. that relied on mass spectrometry to quantify the RT-PCR products [6], the FUSION screen determined gene expression using bead-based (Luminex), branched DNA signal amplification (bDNA) technology (Affymetrix Quantigene 2.0) [12,13,40]. The use of bead-based bDNA technology permits the simultaneous quantification of eight mRNAs in each sample without RNA isolation and can be multiplexed 8-fold during the amplification steps to increase throughput and to decrease reagent costs. Analysis of the microRNA mimic library demonstrated that the six reporter mRNAs had significant variation from each other and could successfully cluster microRNA families that shared seed sequences [13,17]. The gene expression profiles from the microRNA mimic library were combined with those generated by siRNA-mediated knockdown of the 780 genes representing kinase, phosphatase, and similar proteins in the human genome and lead to the identification of two clades of proteins and microRNAs that could modulate autophagy. Both SRMS and BMP2K were biologically validated as novel regulators of autophagy [13].

Targeting genes that are essential drivers of tumorigenesis but are dispensable for normal cell development should lead to therapeutic targets with a high therapeutic index [41]. The Kinase suppressor of Ras 1 (KSR1), a scaffold protein, is a potent modulator of the mitogen-activated protein kinase (MAPK) signaling cascade [42,43,44,45]. KSR1 is essential for Ras-driven oncogenic transformation, but it is dispensable for normal cellular development [46]. Apart from minor defects, *Ksr1*^−/−^ mice are phenotypically and developmentally normal [47,48]. In colorectal cancer (CRC) cells, disruption of KSR1 decreases tumorigenesis in vivo [12]. Although KSR1 is present both in non-transformed and transformed cells, KSR1 is essential to oncogenic Ras-driven cancers. These properties make KSR1 an exciting therapeutic target with a high therapeutic index. Therefore, targeting KSR1 or functionally similar proteins could be lethal to Ras-driven cancer cells with reduced toxicity to normal cells.

In order to identify proteins with functional similarity to KSR1, we developed a gene-expression signature that represents the loss of KSR1 and use it to identify KSR1-like genes by screening 14,355 proteins using an arrayed siRNA library [12,14,40]. These six signature genes demonstrated consistent decrease in expression upon KSR1 depletion and were selected as representative of KSR1 disruption. Reporter gene expression profiles from each unique perturbation were compared to profiles in KSR1 knockdown cells and cells transfected with nontargeting siRNA [40]. The expression data acquired underwent stringent quality control and preprocessing and was normalized to the median gene expression for the nontargeting controls on each plate. The effects on gene expression following each individual gene depletion was calculated using Pearson Correlation (PC) and Euclidean Distance (ED) similarity metrics in comparison to the KSR1 depleted wells after removal of a few siKSR1 outlier wells. Based on the positive control target (siKSR1), the genes were ranked based on the two-similarity metrics. A linear regression cutoff was determined to identify 788 potential target genes in the genome-scale screen that were functionally similar to KSR1 [14,49]. The Euclidean distance and Pearson correlation similarity metrics for individual genes relative to the average signature of KSR1 depletion is available at PubChem (AID: 1259424) [40,50]. The detailed methods describing experimental completion, computation, and bioinformatic analysis are described in published manuscripts [12,40,50]. We used FUSION to identify proteins with functional similarity to KSR1 (Table 1), and three KSR1-like genes are biologically validated hits (AMPKγ1, EPHB4, and TIMELESS) and are discussed below [12,14,18].

AMP-activated protein kinase (AMPK) subunit γ1 (AMPKγ1) was identified through the FUSION Kinome screen as functionally similar to KSR1 [12]. Under conditions of severe stress, AMP binds to the gamma subunit which activates AMPK and promotes signaling of downstream metabolic functions [51]. KSR proteins contain a domain between the conserved areas 2 and 3 (CA2 and CA3) domains that is required for its interaction of AMPK to facilitate signaling. Loss of the CA3 domain by introducing a frameshift and nonsense mutation (E667V and A373T, respectively) disrupts the ability of KSR to bind to AMPK [52,53,54]. AMPK has been shown to promote tumor cell survival by increasing the metabolic capacity of the cell, especially under conditions of stress. Importantly, normal cells were unaffected by the loss of AMPKγ1. The decrease in viability and survival of HCT116 cells following AMPKγ1 depletion was attributed to the decrease in expression of critical metabolic regulator PGC1β [12]. Subsequent analysis determined that AMPKα2, which encodes the kinase domain of AMPK, was also selectively required for survival of colon cancer cells [12,16].

Using FUSION, EPH (erythropoietin-producing hepatocellular carcinoma) receptor B4 (EPHB4) was identified as a KSR1-like, cancer-specific regulator of colon tumor survival and progression [14]. Given its low Euclidian distance and high Pearson correlation coefficient, EPHB4 significantly correlated with the KSR1-depleted signature. These data were validated by knocking down either KSR1, EPHB4, or the nontargeting control with siRNA for 72 h in normal cells or colon cancer cells. The cell viability was unchanged with knockdown of KSR1 or EPHB4 in normal cells compared to the control siRNA and was also found to decrease PGC1β levels via c-myc transcriptional activity. In contrast, KSR1 or EPHB4 knockdown significantly decreased cell viability in colon cancer cell lines HCT116 and Caco2 compared to the control siRNA. Although KSR1 does not have a reliable inhibitor, there is a commercially available inhibitor for EPHB4. The EPHB4 kinase inhibitor, AZ12672857, indeed decreased cancer cell viability, leaving normal cells unharmed. EPHB4 was concluded to be functionally similar to KSR1 and might be targeted successfully in a cancer-specific manner.

Additional analysis of the FUSION screen data identified multiple components of the circadian rhythm pathway as being functionally similar to KSR1 [49]. Among the genes involved in the circadian rhythm pathway, TIMELESS passed all the bioinformatic filters including seed sequence analysis and viability cutoffs [18]. Circadian rhythm proteins are often preferentially expressed in certain portions of a 24-h cycle. This was the case with TIMELESS in normal colon epithelial cells (HCECs). However, in colon cancer cells, TIMELESS is highly overexpressed throughout the entirety of the 24-h cycle. Similarly, in The Cancer Genome Atlas (TCGA) database [55], TIMELESS is significantly upregulated in tumor samples compared to normal tissue. Further validation showed that, in a panel of colon cancer cell lines, TIMELESS mRNA and protein expressions were significantly upregulated compared to HCECs. Cell viability was also markedly reliant on TIMELESS expression in colon cancer cells compared to HCECs, similar to that of KSR1. Further evaluation showed that, in colon cancer HCT116 cells, activated ERK promoted TIMELESS expression. Interestingly, the total mRNA of TIMELESS was unchanged following inhibition of ERK. When polysome-bound mRNA was evaluated, the translation efficiency of TIMELESS was significantly decreased following the ERK inhibitor. Subsequent analysis demonstrated a profound ability of ERK inhibition to shift TIMELESS mRNA from polysomes to monosomes in HCT116 cells (C. Rao, unpublished results). These data suggest that ERK activity in colon cancer cells preferentially promotes TIMELESS translation. In other colon cancer cell lines where ERK inhibition did not affect TIMELESS levels, mTOR inhibition reduced TIMELESS mRNA translation efficiency and protein expression. Lastly, validation of TIMELESS led to the discovery that its role in colon cancer cells was based on cell cycle regulation. Five colon cancer cells, HCT116, HCT15, SW480, SW620, and RKO depleted of TIMELESS had a significant decrease in cell survival and proliferation. Decreased viability was shown to be due to the induction of a G2/M cell cycle arrest via CHK1 phosphorylation, revealing TIMELESS as a potential target for colon cancer therapy, particularly in combination with other DNA damaging agents or CHK1/Wee1 inhibition [18].

### 3.3. Identifying the Method of Action of Uncharacterized Natural Products in Ras-Driven Cancers Using FUSION

Dovetailing siRNA, microRNA, and chemical perturbations using FUSION generated MoA hypotheses for both known small molecules and uncharacterized natural products fractions. Although natural products remain an attractive therapeutic target, a significant hurdle to their use is understanding the biological MoA of these compounds. FUSION matches perturbagens of known mechanism of action (siRNAs, or microRNAs, or characterized small molecules) to those of unknown mechanisms of action (natural products, novel small molecules, or siRNAs targeting genes of unknown function). This high-throughput discovery platform was used to predict MoA for thousands of marine microbe-derived natural products as novel modulators of cellular processes, such autophagy and oncogenic Akt signaling [13,17,20]. Mapping of uncharacterized natural product fractions libraries to microRNA–siRNA clusters known to regulate autophagy, such as ULK-1 and mTOR clusters, identified small molecules isolated from the *Streptomyces bacillaris* strain SN-B-019. The natural product fractions that correlated with reporter gene signature with ULK1 knockdown were further isolated and characterized. Characterization of the pure form of compound, SN-B-019-cmp1, was shown to induce the accumulation of ULK-1 downstream target LC3 and blocked maturation of autophagosomes, and its structure was revealed to be bafilomycin D [13].

Furthermore, analyzing the FUSION network map of natural products targeting AKT signaling identified the natural product fraction SN-A-024, produced from the *Streptomyces coeruleoaurantiacus* strain [20]. The natural product fraction clustered with siRNA clusters of two AKT regulatory proteins, TBK1 and PDPK1. Structural characterization of the biologically active metabolite of the compound revealed N^6^,N^6^-dimethyladenosine. Consistent with the initial analysis of SN-A-024, N^6^,N^6^-dimethyladenosine rapidly decreased the AKT activity in various non-small cell lung cancer (NSCLC) cell lines HCC44, H322, H2009, H595, and H1993 [20]. McMillan et al. performed a genome-wide integration of siRNA targeting distinct genes, micro-RNA perturbations, and marine-derived natural product and chemicals to generate a comprehensive network of similarity maps [17]. Among the various clustering of natural products with similar function, SN-A-022-6 was validated to be mechanistically similar to a potent ERRα inhibitor, XCT790 [17]. SN-A-022-6 not only behaved similar to XCT790 to induce stress induced mitochondrial damage but also was shown to induce dose-dependent suppression of oxygen consumption.

Since AMPKγ1 was demonstrated to be selectively required for colon tumor cell survival, a gene expression signature was generated in FUSION to identify small molecule inhibitors that mimic the effects of AMPK inhibition. Compound C or dorsomorphin is a known nonspecific competitive AMPK inhibitor, using the gene expression signature induced by this molecule as a reference for comparison to signatures from the natural product screen [16]. FUSION identified several fractions isolated from *Streptomyces bacillaris*, SN-B-004, clustered with Compound C. Out of nineteen SN-B-004 fractions, treatment with fractions 13–17 inhibited downstream targets of AMPK and decreased viability of HCT116 cells. SN-B-004 fractions 13–17 were shown to be mechanistically and functionally different from Compound C and were subsequently identified as 5′-hydroxy-staurosporine (5-OH-S) using mass spectrometry and nuclear magnetic resonance spectroscopy. Upon further evaluation, 5-OH-S was shown to inhibit AMPK kinase activity and downstream signaling. As expected, 5-OH-S was also shown to decrease cell viability of colon cancer cells, HCT116 and SW480, but not normal cells, supporting the concept that FUSION is an effective method for screening new drugs while simultaneously identifying their targets.

These analyses are a proof-of-concept validation for detection of functionally related proteins and small molecules using FUSION. Using a reference standard, in this case KSR1, that is required for the survival and transformed properties of tumor cells but not non-transformed cells from the cognate tissue of origin allows FUSION to specifically detect cancer-specific targets.

## 4. Guidelines for Designing, Performing, and Analyzing Future FUSION Screens

Challenges associated with FUSION can be largely attributed to limitations on throughput imposed by the workflow. RNA quantification can be a persistent bottleneck, which limits the number of samples that can be processed simultaneously, making the screen vulnerable to significant variation and batch effects. Optimizing the screen to minimize these batch effects and variation is critical in order to identify hits that are biologically valid. Technical constraints on the number of genes that can be multiplexed also limits the amount of biological space on which the assay can report. The cell model chosen for screening imposes inherent limitations on which signaling pathways can be interrogated, even if the perturbation library is genome wide. If a signaling pathway is not active in the cell line, perturbagens against that pathway will not affect gene expression readout. Thus, although the FUSION assay is performed in a disease- and tissue-agnostic manner, the results are nonetheless biased by the screening context that is chosen. Here, we discuss strategies to optimize the protocol and data processing and to minimize the effects of variation in FUSION.

### 4.1. Batch Effects and Normalization

As with any screen in which bottlenecks in the sample processing arise, detection and correction of batch effects are essential for effective analysis of FUSION data. Neilsen et al. found that the geometric mean of the housekeeping genes varied the most when cells were cultured and transfected at different times [40,49]. This variability was likely due to cell culture density as the raw value of housekeeping genes is a surrogate for cell number. Samples that were transfected simultaneously but assayed on different dates did not demonstrate appreciable batch effects [40]. Normalization between batches transfected on different dates minimizes this variation. Ideally, this normalization is performed on a per well basis by measuring multiple housekeeping genes in every well and across plates by using additional multiple control wells present on every plate.

Selection of an appropriate method for normalization is critical to the generation of robust data. Traditional statistical methods include normalizing to plate position based on the average or median reading of all plates for a given well position or normalizing to average or median readings within a given plate or batch, which has been previously used for FUSION [13,17]. These strategies rely on basic statistical assumptions including normal distribution, random sampling, and equality of variance. Due to the potential for non-random plating and batch effects, the analysis of any screen data, including that from FUSION, presents some unique challenges. For example, siRNA libraries are often clustered by function. Some plates may be enriched relative to others if certain categories of genes are more likely to affect gene expression, such as genes from a kinome group. If experimental factors increase the scale of measurements between plates or batches, normalization to reporter median will be skewed to select or reject hits from plates affected by those factors. Ideally, automatic plating machinery would be programmed to localize plate controls in different positions on each plate and to plate siRNAs from the library in random order. However, this usually is not feasible due to the fixed position of reagents in arrayed libraries and most automated pipetting machinery. In these cases, normalization to plate position or plate/batch median may not be appropriate.

An alternative method for libraries not plated randomly involves normalizing expression of reporter genes in each well to a nontargeting siRNA negative control on the same plate (negative control normalization). This normalization is performed after normalizing to housekeeping gene expression and subtracting backgrounds measured from blank well controls. When controls are included on every plate, negative control normalization is appropriate for large data sets that may have considerable variation between batches. Importantly, this method does not rely on the statistical assumptions discussed previously and instead depends on the biological assumption that the ratio of gene expression in negative to positive controls stays constant.

After normalizing the six reporter genes to the geometric mean of the two housekeeping genes, we performed statistical comparison of reporter median, plate position, and negative control normalization using data from the multiple KSR1-depleted wells present on each plate to evaluate accuracy, precision, and scalability. Both reporter median and plate position normalization showed low accuracy, as measured by their ability to rank hits that have been biologically validated to be KSR1-like, among the top 5% of results. Reporter median normalization displayed high precision but low scalability (r = 0.66), whereas plate position normalization displayed high scalability (r = 0.99) but low precision between KSR1 depleted wells. The negative control normalization method was found to be superior to plate position or reporter median normalization in all three parameters. Precision and scalability of this method are depicted in Figure 1.

Lastly, in FUSION, additional controls to monitor transfection efficiency were added to each plate, including an siRNA targeting the housekeeping gene PPIB and a lethal control targeting PLK1, and were found to be over 95% on every plate [40].

### 4.2. Minimizing Cost

A major challenge associated with FUSION resides in the cost and complexity of the screen. Additionally, substantial effort is also required to complete the computational analysis and biological validation of a potential target of a small molecule, which can often be time-consuming and expensive. However, when the goal is to identify potential targets for a given small molecule or novel molecules for a specific target, the cost of FUSION may be more accurately evaluated relative to the investment required for identification of targets and molecules by more traditional methods. Modifications that increase robustness of data often also increase cost as well as time and complexity of the experiment. For example, conducting an analysis in several different cell lines increases the likelihood of selecting hits that demonstrate biological validation and increases generalizability of results but also multiplies the cost of the screen. Additionally, increasing the number of reporter probes or number of housekeeping genes potentially increases specificity of the signature; however, this must be weighed against increased cost as well as technical limitations. The addition of more housekeeping genes increases resolution only if the information obtained is independent from other probes and are not covariant. We suggest, where possible, to select probes with both increased expression and decreased expression in response to the reference perturbation. Probes should also be analyzed for covariation since this affects the validity of Euclidian distance in the similarity analysis. If covariation is found, the Mahalanobis distance can be used instead [3]. Note, however, that while this concern can be mitigated through the application of Mahalanobis distance, covarying probes add no additional resolution to the analysis while substantially increasing the cost, but the true variation can only be assessed after the screen is performed.

Next-generation versions of FUSION incorporate these concepts while balancing costs. In this iteration, multiple NSCLC cell lines are assayed with a larger reporter probe set tuned to survey the biological space in this tissue setting (M. White, J. MacMillan, personal communication). This tuning is achieved by selecting reporter probes with maximal variance among a large panel of NSCLC cell lines and in response to perturbation and without covariation. In addition, cell lines have been chosen for their ability to best represent the baseline expression profiles of other NSCLC cell lines, ensuring maximum generalizability of results. Limiting the number of cell lines and number of probes used can minimize cost while increasing assay sensitivity and resolution.

### 4.3. Off-Target Effects and False Positives

RNAi is known to induce off-target effects, which can lead to false positives in screening. Of note are microRNA-like seed sequence effects. microRNAs regulate protein expression by binding to mRNAs to inhibit translation, and this binding can occur even when the microRNA and mRNA sequences are slightly mismatched. While RNAi depends on complementarity to induce target mRNA degradation, siRNA has also been shown to bind off-target to imperfectly matched mRNAs inducing microRNA-like downregulation [56]. This effect appears to be most commonly mediated by the first several bases of the siRNA, referred to as the seed sequence [57]. Several bioinformatic methods have been developed to detect the potential effects of RNAi seed sequences [58,59,60]. These methods rely on analyzing siRNA/shRNA hits for enrichment in a common subsequence to detect microRNA-like off-target effects common to the RNAs that could be mediating gene expression changes. These methods can filter false positives from candidate genes prior to more resource-intensive biological validation.

To provide further evidence that gene expression phenotype is due to knockdown of the target protein rather than off-target knockdowns, a follow-up screen can be performed on wells selected as hits from the initial, arrayed screen. This secondary screen involves deconvolution of pooled siRNA such that a single siRNA is plated into individual wells or a new set of unrelated siRNAs. This allows for analysis of the phenotype in the individual siRNA wells to confirm phenotypic concordance between the pooled siRNAs and individual siRNA knockdowns. This avoids a scenario where a single siRNA off-target effect falsely induces the phenotype of interest.

FUSION-based screening can also lead to “false positive” MoA predictions due to inherent limitations on resolution in the reporter gene set chosen. Some perturbations with disparate MoAs could produce highly similar expression signatures if the genes being assayed are not optimized to report on or discriminate between the involved pathways. Thus, follow-up experimentation to validate predictions are important, as is careful selection of reporter genes, to maximize the resolution of the assay.

## 5. The Future of FUSION

Recent developments in FUSION data analysis include the application of Similarity Network Fusion (SNF) to integrate with other datasets [19]. SNF was first applied to the integration of microRNA, mRNA, and DNA methylation profiling in order to identify novel glioblastoma subtypes [61] and has since been successfully used to integrate orthogonal datasets in other contexts [62,63]. Integration of FUSION data with orthogonal screening platforms has the potential to increase the biological space covered while mitigating limitations present in analysis of an individual dataset.

In addition to new developments in FUSION data analysis, improved understanding of how culture conditions affect gene expression should also be considered in future FUSION screens. HTS carried out on cultured cells propagated on two-dimensional (2D) plastic surfaces are not representative of cells residing in a complex tissue microenvironment, which is one of the contributors to the high failure of cancer drugs entering clinical trials [64,65]. Extracellular matrix and other microenvironment factors influence the cell response to drug efficacy by either altering the drug’s MoA or by promoting drug resistance [65,66]. Therefore, three-dimensional (3-D) technologies are now being pursued to accommodate higher precision in drug discovery. In vitro 3D cell culture models more closely resemble tumors in patients and can be used for studying drug response in high-throughput genetic screen [67,68]. A genome-wide CRISPR screen in lung cancer cells grown as 2D monolayers and 3D spheroids showed that 3D culture more closely mimicked cancer dependencies seen in patients [68]. In vitro 3D organoid models alleviate some of the issues of 2D models, more closely resemble tumors in patients, and can be used for studying drug response in high-throughput genetic screens [67,68]. The heterogeneous nature of the organoids increases the complexity and may decrease the statistical power compared to screens in cell lines. However, 3D screens have a major advantage since 3D cultures can serve as the last step of in vitro testing just prior to testing the drug response in vivo and are likely in vivo to identify valuable targets for personalization and identification of cancer vulnerabilities in vivo. Conducting FUSION in 3D, either by using anchorage independent growth or in organoid culture, could increase applicability and generalizability of future screens, such that screened hits are more likely to be valid across cell lines and in vivo.

## 6. Conclusions

The development of high-throughput screening techniques has been an undeniable advancement in the cancer research field. The ability to identify novel genes essential to cancer cell survival and proliferation has presented researchers with several new targets for drug development. Unfortunately, many of the genes identified by the screens are poor drug targets due to cellular location or other factors. We have demonstrated that FUSION screening can be used to identify functionally similar genes which, importantly, may be targetable and equally important to the cell.

Novel developments that can improve FUSION screening include CRISPR/Cas9 genome editing for an expanded scope of testable perturbations as well as enhanced data analysis using integrative approaches. While batch effects and plate variation are still a concern, selection of appropriate normalization methods and careful experimental workflow design can help minimize these effects. Furthermore, machine learning algorithms are emerging as a method to optimize selection of cell lines and reporter probes in order to minimize costs. Applying 3D culture techniques has the potential to further expand the applicability and generalization of FUSION to an even more biologically relevant screen.

## Figures and Tables

**Figure 1 cancers-12-03143-f001:**
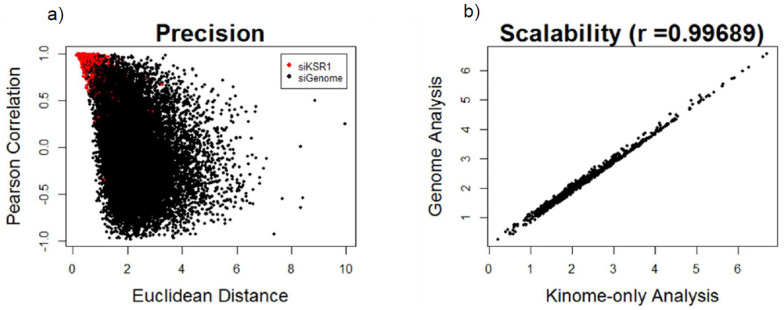
Evaluation of negative control normalization based on precision and scalability: (**a**) scatterplot of Kinase suppressor of Ras 1 (KSR1)-depleted wells (red dots) and individual gene depletions from the siGenome library (black dots) based on Euclidean distance (arbitrary distance units) and Pearson correlation similarity metrics and (**b**) scatterplot showing correlation of the Euclidean distance metrics after normalizing to the nontargeting control wells using the ranking of previously validated kinome hits calculated from the kinome only analysis (*x*-axis) vs. the entire-genome screen analysis (*y*-axis).

**Table 1 cancers-12-03143-t001:** The summary of novel findings identified using Functional Signature Ontology (FUSION).

Screen Strategy	Key Findings	Identified NP/Protein	Proposed Mechanism	Reference
**Functionally related proteins to KSR1**	AMPKγ1 as an essential driver of PGC1β expression and colon cancer cell survival.	AMPKγ	α2β2γ1 isoform of AMPK promotes aberrant expression of tumor-specific PGC1β and ERRα expression in colon cancer.	Fisher and Das et al. [12]
	EPHB4 supports colon cancer cell survival through regulation of Myc and PGC1β mRNA levels.	EPHB4	KSR1 promotes EPHB4 expression by protecting it from lysosome-dependent degradation. EPHB4 kinase inhibitor AZ12672857 is selectively toxic to colon cancer cells.	McCall and Gehring et al. [14]
	Increased ERK signaling through oncogenic Ras promotes TIMELESS overexpression in cancer promoting cancer cell proliferation.	TIMELESS	TIMELESS depletion induces cell cycle checkpoint-induced G2/M arrest limiting cell proliferation.	Neilsen and Frodyma et al. [18]
**Natural product fractions**	SRMS, BMP2K, and natural products SN-B-019-cmp1 as autophagy modulators.	Isolated from *Streptomyces bacillaris* strain, SN-B-019-cmp1	SN-B-019-cmp1 and bafilomycin D block autophagasome maturation. BMP2K blocks basal autophagy, while SRMS functions through mTOR to inhibit autophagy.	Potts, Kim, and Fisher et al. [13]
	SN-B-004 and 5′-hydroxy-staurosporine (5-OH-S) as a novel AMPK inhibitor.	*Streptomyces bacillaris* strain SN-B-004	SN-B-004 and 5-OH-S are competitive AMPK inhibitors and decreased levels of AMPK downstream targets ACC and Raptor. AMPK inhibition via 5-OH-S is selectively toxic to colon cancer cells.	Das and Neilsen et al. [16]
	De novo network of targets of uncharacterized natural products.	SN-A-022-6	SN-A-022-6 clustered with ERRα inhibitor XCT790 and suppressed mitochondrial oxidative capacity in lung cancer cells.	McMillan and Kwon et al. [17]
	Marine-derived natural product as a potent AKT inhibitor.	N^6^,N^6^-dimethyladenosine	SN-A-024-A and N^6^,N^6^-dimethyladenosine behave as small molecule inhibitors of AKT signaling.	Vaden et al. [20]

NP, natural product.

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
