# Peer review of "A Gene Expression High-Throughput Screen (GE-HTS) for Coordinated Detection of Functionally Similar Effectors in Cancer"

_cancers, 2020, doi:10.3390/cancers12113143_

Round 1

Reviewer 1 Report

The manuscript is very well written and clear.  The authors noted the issues with normalization.  The inherent issue of using only two housekeeping for normalization was not directly addressed and it was not entirely clear in the section 4.1 of the author's conclusions about the current approaches or what was recommended as best practices.  There is an extensive literature on the normalization issues in the RNAseq literature.  e.g see RUVg methods.  I also wonder if it would be helpful to have some discussion/comparison with other similar integrative/reductive approaches that are being utilized to make sense of high throughput data.

Reviewer 2 Report

The review article by Rao and colleagues (manuscript cancers-965209) examines the use of Functional Signature Ontology (FUSION), a gene expression-based HTS method that heavily relies on bioinformatics and biostatistics, to determine pathway targets for small molecules and microRNA mimics and to identify genes with functional similarity to a target gene.

The authors review the advantages of FUSION over single endpoint phenotypic screens to identify hits from chemical libraries. They discuss screening strategies to identify functional similar proteins in Ras-driven colorectal cancer (CRC). As Ras is still undruggable, functional similar proteins with known inhibitors can overcome this bottleneck in Ras-driven cancers. They discuss their own siRNA screen on the Ras/MAPK scaffold KSR1. The review ends with a discussion on guidelines to optimize the design and analysis of future FUSION screens and how integrating orthogonal screening platforms and replacing 2D cell cultures with organoids will improve HTS screen outcomes.

The review is concise with a clear message. It is well-written with the average expert or new researcher to the topic in mind. It is therefore expected to reach a wide audience. The key concepts are well explained even though some of the terminologies are not.

A few points that could clarify the message:

  • Line 223: EPH as ephrin the first time it is introduced.
  • Line 247: the last sentence at the bottom of page 5 ‘When polysome-bound mRNA’ is truncated and needs to be completed. I believe some text is missing.
  • The identification of the circadian rhythm protein TIMELESS as functionally similar to KSR1 is very interesting. Can the authors comment on this finding especially that TIMELESS is not the only gene regulated by ERK or upregulated in CRC.
  • The KSR1 screen was done in one cell line HCT116 and there was no mention of validation in additional colon cancer cells or other KRAS-transformed cells. The authors should not be quick to generalize their conclusions.
  • Table 1: follow the text flow and start with KSR1 functionally related proteins and then move to natural product fractions. Also, under Key Findings, the short sentences for AMPKg, EPHB4, and TIMELESS need to be improved.
  • Line 276, start a new paragraph as is the case on line 289.
  • Line 285 is not clear and needs editing.
  • The paragraph on Dorsomorphin (lines 289-299) needs clarification and editing. What is meant by fractions 13-17?
  • In Figure 1a, what is the unit of the Euclidean distance on the horizontal axis. Explain what 2 or 10 units mean. Figure 1b, what do the X- and Y-axis units represent? This figure is probably clear for the authors but not for the non-expert reader.
  • Line 363, under “Minimizing the Cost”, I agree with the authors that FUSION screen is complex and expensive, but the cost has to be compared to the time and cost of identifying the real target of a small molecule, which can be long and very expensive.
